# Chromogenic *LMO2* mRNA ISH Expression Correlates with LMO2 Protein and Gene Expression and Captures Their Survival Impact in Diffuse Large B-Cell Lymphoma, NOS

**DOI:** 10.3390/cancers16132378

**Published:** 2024-06-28

**Authors:** Natalia Papaleo, Andrea Molina-Alvarez, Ricard Onieva, Diana Fuertes, Blanca Sanchez-Gonzalez, Xenia Riera, David Lopez-Segura, Carmen Lome-Maldonado, Xavier Ara-Mancebo, Jose Yelamos, Marta Salido, Ivonne Vazquez, Xavier Calvo, Luis Colomo

**Affiliations:** 1Department of Pathology, Hospital del Mar, Hospital del Mar Research Institute-IMIM, 08003 Barcelona, Spain; nfpapaleo@tauli.cat (N.P.); 66525@psmar.cat (A.M.-A.); 62527@psmar.cat (X.R.); dlopezsegura@psmar.cat (D.L.-S.); 40621@psmar.cat (C.L.-M.); 68606@psmar.cat (X.A.-M.); jyelamos@psmar.cat (J.Y.); msalido@psmar.cat (M.S.); ivazquez@psmar.cat (I.V.); xcalvo@psmar.cat (X.C.); 2Department of Pathology, Consorci Hospitalari Parc Tauli, Institut d’Investigació i Innovació Parc Taulí (I3PT), 08208 Sabadell, Spain; ronievac@tauli.cat; 3Department of Morphological Sciences, Universitat Autonoma de Barcelona, 08193 Barcelona, Spain; 4Department of Health and Experimental Sciences, Universitat Pompeu Fabra, 08003 Barcelona, Spain; 5Research Unit Support, Institut d’Investigació i Innovació Parc Taulí (I3PT), 08208 Sabadell, Spain; dfuertes@tauli.cat; 6Department of Hematology, Hospital del Mar, Hospital del Mar Research Institute-IMIM, 08003 Barcelona, Spain; bsanchezgonzalez@psmar.cat

**Keywords:** LMO2, MYC, CISH, large B-cell lymphoma

## Abstract

**Simple Summary:**

In this study, our primary aim was to evaluate *LMO2* mRNA expression in tissue, an area with limited research. Studies assessing *LMO2* mRNA expression overlook tissue morphology since they have relied on microarray platforms or PCR methods. Our results show a reliable method to identify *LMO2* mRNA expression in tissues, capturing the reported biologic features of LMO2, that may be useful to study *LMO2* mRNA expression in hematological neoplasms and solid tumors.

**Abstract:**

Background: *LMO2* is a relevant gene involved in B-cell ontogeny and a survival predictor of aggressive large B-cell lymphomas (aLBCL). Most studies assessing *LMO2* mRNA expression have relied on microarray platforms or qRT-PCR methods, overlooking tissue morphology. In this study, we evaluate *LMO2* RNA expression by chromogenic in situ hybridization (CISH) in normal tissue and in a series of 82 aLBCL. Methods: *LMO2* CISH was performed in formalin-fixed paraffin-embedded tissues, scored by three different methods, and correlated with a transcriptome panel. Results: We obtained statistically significant results correlating the methods of evaluation with LMO2 protein expression and gene expression results. Normal tonsil tissue showed high levels of *LMO2*, particularly within the light zone of the germinal center. Conversely, in aLBCL, a notable reduction in *LMO2* expression was noted, remarkably in cases carrying *MYC* rearrangements. Furthermore, significant results were obtained through overall survival and Cox regression survival analysis, incorporating International Prognostic Index data alongside *LMO2* expression levels. Conclusions: We show a reliable method to identify *LMO2* mRNA expression by CISH, effectively capturing many of the reported biologic features of LMO2.

## 1. Introduction

LIM-domain only 2 (LMO2) is a cysteine-rich LIM domain transcription factor located on the short arm of human chromosome 11 (11p13). First described as a chromosomal translocation oncogene partner of the TCR loci in a subset of T-cell acute lymphoblastic leukemia (T-ALL) [1,2], subsequent studies in murine models revealed its involvement in hematopoiesis and blood vessel formation [3]. Gene expression profiling (GEP) identified *LMO2* as one of the genes associated with the germinal center (GC) signature in secondary follicles and the GCB-like profile in diffuse large B-cell lymphomas (DLBCL). The survival analyses demonstrated a significantly higher survival rate for patients with GCB-like DLBCL in both pre-rituximab and rituximab eras compared to those with activated B-like DLBCL, which are characterized by the expression of genes typically induced during in vitro activation of peripheral blood B-cells [4,5,6]. Further studies translating GEP findings into clinical practice consistently identified *LMO2* as one of the most significant prognostic genes in DLBCL [7,8]. Furthermore, several studies have shown that LMO2 protein expression showed a significant favorable prognostic impact in DLBCL-NOS treated with immunochemotherapy [9,10,11].

Most RNA-based studies used microarray platforms or quantitative real-time polymerase-chain-reaction analyses in fresh frozen or formalin-fixed paraffin-embedded (FFPE) samples to evaluate *LMO2* gene expression, and therefore tissue morphology was not preserved. In the present study, we evaluate, for the first time, *LMO2* mRNA expression using chromogenic in situ hybridization (CISH) in both normal tissues and a series of aggressive large B-cell lymphomas (aLBCL). We used a CISH assay, which allows high amplification of mRNA signals identifiable by a common bright-field microscope. Each individual RNA molecule was analyzed as a distinctive dot of chromogenic precipitate in each cell. The results were correlated with LMO2 and MYC protein expression as well as *MYC* gene status. Furthermore, GEP performed in a subset of cases provided useful information to validate the results, and survival analyses were evaluated in a cohort of DLBCL-NOS patients treated with immunochemotherapy.

## 2. Materials and Methods

### 2.1. Case Selection

All FFPE tissue samples were obtained from the archive of the Department of Pathology, Hospital del Mar, Barcelona, based on material availability, between 2000 and 2019. The cases included were aLBCL received at our institution both as diagnostic consultation and to perform cytogenetic studies and were diagnosed following the criteria of the International Consensus Classification (ICC) and the 5th edition of the WHO classification (WHO-HAEM5) [12,13].

We studied a series of 82 aLBCL, including 13 (16%) transformed DLCBL (tDLBCL; 1 marginal zone lymphoma-MZL, 12 follicular lymphomas-FL), 56 (69%) DLBCL-NOS, 9 (11%) high-grade B-cell lymphomas with *MYC* and *BCL2* rearrangements), and high-grade B-cell lymphomas with *MYC* and *BCL6* rearrangements that we have abbreviated as HGBL-DH/TH (1TH, 6 HGBL *MYC/BCL2*, and 2 HGBL *MYC/BCL6*), 2 (2%) HGBL-NOS, and 2 (2%) BL, both in HIV-positive patients. Primary mediastinal large B-cell lymphoma, primary central nervous system lymphoma, primary cutaneous diffuse large B-cell lymphoma, leg type DLBCL, T-cell/histiocyte-rich B-cell lymphoma, HHV8-associated lymphoproliferative disorders, plasmablastic lymphoma, and transformed myeloma were excluded. The Ethics Committee of the Hospital del Mar of Barcelona (2017/7481/I) approved the study. Informed consent to use both clinical data and histological material was obtained in accordance with the Declaration of Helsinki. 

For survival analysis purposes, clinical data (including IPI, gender, response to treatment, and follow-up) of the patients were collected and analyzed. Forty-three patients diagnosed with DLBCL-NOS and treated with immunochemotherapy (4 R-CVP, rituximab plus cyclophosphamide, doxorubicin, vincristine, and prednisone; 35 R-CHOP, rituximab plus cyclophosphamide, doxorubicin, vincristine, and prednisone or prednisolone; 4 R-EPOCH, etoposide, prednisone, vincristine, cyclophosphamide, doxorubicin, and rituximab) were analyzed to evaluate the survival outcomes of *LMO2* CISH.

### 2.2. Immunohistochemistry

A common panel of B- and T-cell immunohistochemistry (IHC) markers in WTS was used for diagnosis. Tissue fixation (10% buffered formalin) and processing were performed following standard methods. Immunohistochemical stains were carried out for each case with monoclonal and polyclonal antibodies reactive in paraffin-embedded tissue sections using fully automated protocols on a BenchMark XT and Ultra immunostainers (Ventana Medical Systems, Tucson, AZ, USA). The antibodies used included common B- and T-cell markers [CD19 (clone SP110), CD20 (clone L26), CD79a (clone SP18), CD3 (clone 2GV6), CD5 (clone SP19), BCL2 (clone 124), CD10 (clone SP67), BCL6 (clone GI191E-A8), MUM1/IRF4 (clone MRQ-43), Ki-67 (clone MIB-1), LMO2 (clone SP51), and MYC (clone Y69), following the manufacturer’s instructions. The conditions and the evaluation for all these antibodies were the same as previously described and were assessed as previously described, using appropriate internal and external controls [11,14]. Cell of origin (COO) was assigned by using the Hans immunohistochemistry algorithm [15]. 

### 2.3. Fluorescence and Chromogenic In Situ Hybridization (FISH and CISH)

Fluorescent ISH (FISH) studies were performed using dual color break-apart probes for *MYC* and *BCL6*, and dual color dual fusion probes for *IGH/BCL2*, (all provided by Vysis, Abbott Molecular, Des Plainescity, IL, USA). FISH was performed and evaluated as described previously following the criteria of Ventura [16].

The chromogenic ISH (CISH) study was performed for *LMO2* and *MYC* genes with RNAscope using the 2.5 Assay for the Ventana Discovery Ultra system (Advanced Cell Diagnostics—ACD, Hayward, CA, USA). The *LMO2* probe was customized by the manufacturer, which designed and provided the probe Hs-LMO2-C1 Homo sapiens LIM domain only 2 (*LMO2*) transcript variant 1 mRNA, NM_005574.4. (#1029259-C1); the *MYC* probe was commercially available (#311769, NM_002467.4). The assay was performed as described, using, in parallel, positive and negative controls [17]. The slides were scanned using the Ventana DP200 scanner at 40×. All cases were assessed by two pathologists (NP, LC) using conventional optical microscopy and by the Aperio ImageScope slide viewing software v12.1 (Leica Biosystems, Wetzler, Germany). 

CISH studies were performed in 3 TMAs, including 2 representative 1mm cores of each case being constructed. Whole-tissue sections (WTS) from 27 cases were also included and evaluated for *LMO2* and *MYC* CISH. For each TMA, a hematoxylin-eosin was obtained and used as a control to evaluate the neoplastic areas.

The evaluation of CISH was performed following 3 different approaches provided by the manufacturer as shown in Figure 1 (https://acdbio.com/services/quantitative-analysis (accessed on 25 June 2024)). First, the semi-quantitative visual score (SQV) was based on the predominant staining pattern seen throughout the entire sample or within a defined region of interest, counting the number of dots per cell. Categories were as follows: score 0, 0 dots in 0% cells, score 1, 1–3 dots/cell in 1–25% cells; score 2, 4–9 dots/cell in 26–50% cells; score 3, 10–15 dots/cell in 51–75% cells; score 4, >15 dots/cell in >75% cells. The second approach, which we designated the ACD score, added to the previous approach the evaluation of clusters of dots. Then, the presence of few clusters or cases with 4–9 dots/cell was score 2, less than 10% of clusters and/or cases with 10–15 dots/cell was score 3, and more than 10% of clusters and/or more than 15 dots/cell was score 4. Finally, a weighted H-score across the whole sample was performed. Cells were grouped into 5 bins based on the number of dots per cell as in SQV scoring, and the percentage of cells in each bin was scored according to the following weighted formula: H-Score = 0 × (% of cells in bin 0) + 1 × (% of cells in bin 1) + 2 × (% of cells in bin 2) + 3 × (% of cells in bin 3) + 4 × (% of cells in bin 4). 

We studied the correlations between the three methods, and the results were correlated with LMO2 and MYC protein expression as well. Additionally, GEP performed in a subset of cases provided useful information to validate the results.

### 2.4. Gene Expression Analysis

In fifteen cases, whole transcriptome sequencing was performed on archival FFPE. All samples were studied using the HTG EdgeSeq system with the HTG EdgeSeq Transcriptome Panel (HTG Molecular Diagnostics, Inc., Tucson, AZ, USA, #916-001) at the IVO Molecular Biology Laboratory (Valencia, Spain), according to the manufacturer’s protocols. This assay contains probes for 19616 mRNA transcripts, including negative and positive process controls. Briefly, after sample lysis, a quantitative nuclease protection assay was performed enabling mRNA quantitation in an extraction-free format. The library was prepared with the addition of tags (barcodes), sequencing adaptors, and tags in a thermocycler. The labeled DNA protection probes were concentrated, pooled, and sequenced using standard NGS protocols on the Illumina NextSeq 550 instrument. Finally, initially demultiplexed FASTQ files were provided as an Excel file.

The 15 cases included occurred in 9 men and 6 women, with a median age of 63.3 years (range 38–82 years). Twelve cases (75%) were nodal and 3 extranodal. The diagnoses were as follows: 5 tFL, 7 DLBCL-NOS, 2 HGBL-DH/TH and 1 BL. 

### 2.5. Statistical Analysis

Data were compared using the Chi-square test, unpaired *t*-tests, or nonparametric tests when necessary; *p* values < 0.05 were considered statistically significant for all tests. Definitions of complete response, progression-free survival, and overall survival were used, and survival analyses were carried out according to the method described by Kaplan and Meier, and the curves were compared by the log-rank test. Multivariate models for survival were performed using the stepwise proportional hazards model (Cox). Multivariate analysis was carried out with R studio 2023.03.0 software, R version 4.3.0 (2023-04-21 ucrt), and SPSS statistics v28 (IBM Corp., Armonk, NY, USA). 

## 3. Results

The whole series included 44 males and 38 females, with a mean age of 67 years (range 38–93). Table 1 shows the descriptive clinicopathological features of the series. 

Across the cohort, high-intermediate/high IPI scores were observed in 13 out of 15 cases with *MYC* rearrangements (*MYCr)* for which information was available (*p* = 0.01). Furthermore, IPI scores correlated with LMO2 and MYC protein expression, since 22 out of 30 (73%) patients with high-intermediate/high IPI scores were LMO2 negative, and 21/30 overexpressed MYC protein (*p* = 0.01 and *p* = 0.019, respectively). Additionally, 14 of 20 *MYCr* cases were LMO2 negative by IHC (*p* = 0.004).

Figure 2 shows the correlations between *LMO2* and *MYC* CISH regarding GEP cases, CISH scoring systems, protein expression, and *MYC* FISH. All LMO2-related results showed significant associations between them. Using non-parametric tests, the comparison of the three algorithms showed strong significant results (*p* < 0.005 for all results), indicating the utility of these three methods to evaluate *LMO2* ISH. In addition, the comparisons between *LMO2* GEP and the scoring methods were strongly significant (*p* < 0.005 comparing *LMO2* GEP vs. ACD and SQV scores; *LMO2* GEP and H-score, *p* = 0.002), validating their utility. LMO2 IHC was also associated with GEP and scoring systems (*p* < 0.005, for all tests). 

Moreover, all MYC-related results were also statistically associated. Therefore, the three algorithms had significant associations between them (*p* < 0.005, for all tests), as well as *MYC* GEP (*p* < 0.005, comparing *MYC* GEP vs. ACD and SQV scores; *p* = 0.03 between *MYC* GEP and *MYC* H-score), MYC protein expression (*p* < 0.005 MYC IHC vs. *MYC* H-score; *p* = 0.01 and *p* = 0.05 for MYC IHC vs. *MYC* ACD and SQV scores, respectively), and *MYC* FISH (*p* < 0.005, for all tests). *MYC* FISH, in addition, was associated with *LMO2* H-score (*p* < 0.005).

Table 2 and Figure 3 show the pattern of expression of *LMO2* and *MYC* in non-neoplastic tonsil tissue. The values were obtained after counting five germinal centers (GC) and their adjacent structures. Similar to LMO2 protein, *LMO2* expression was notably high within the GC, although with distinct zonation, primarily concentrated in the light zone (LZ). Lower levels of *LMO2* in positive cells were observed in the mantle/marginal zones and interfollicular areas, while endothelial cells and intraepithelial lymphocytes exhibited elevated *LMO2* expression. *MYC* ISH expression was predominantly localized in the LZ of the GC, mirroring its protein expression pattern [18,19]. In this area, cells displayed a high number of dots (between 5 and 15), but not dense clusters. Conversely, the dark zone (DZ), mantle/marginal zones, and interfollicular areas showed fewer positive cells. Additionally, high *MYC* expression was observed in the lower layers of the squamous epithelium of the tonsil, consistent with MYC protein expression.

Table 2 also shows the median values of the H-scores for *LMO2* and *MYC* ISH in all cases and by histological subtypes. Compared with the results observed in the normal GC, *LMO2* expression decreased remarkably in all aLBL types. As observed in GEP studies, cases carrying *MYCr* had a significantly lower expression of *LMO2* mRNA (median *LMO2* H-index, 5 for 20 cases carrying *MYCr* vs. *LMO2* H-index, 35 for 62 non-*MYCr* cases, *p* = 0.01). Significant differences were observed for tFL carrying *MYCr* compared with non-rearranged cases (*LMO2* H-index of 40 for 5 cases with *MYCr* vs. 120 in 7 cases non-*MYCr*, *p* = 0.005), and a trend for DLBCL-NOS (*LMO2* H-index of 5 in 2 *MYCr* DLBCL vs. 15 in 54 non-*MYCr* DLBCL-NOS, *p* = 0.11). 

*MYC* H-scores were elevated in BL, HGBL-NOS, and HGBL DH/TH cases. Five tFL with *MYCr* had a high median *MYC* H-score of 170, and 2 DLBCL-NOS acquiring *MYCr* had *MYC* H-scores of 150 and 5. Among 62 aLBCL non-*MYCr*, only 3 cases (5%) had *MYC* H-scores over 100 (1 tFL with gains of MYC; 1 tMZL and 1 DLBCL-NOS with no *MYC* alterations). Table 3 summarizes the results in *MYCr* and non-*MYCr* cases. Regarding cases with gains of *MYC* identified by FISH, they occurred in 4 tFL (median *MYC* H-score, 70) and 12 DLBCL-NOS (median *MYC* H-score, 30). 

Among 43 patients diagnosed with DLBCL-NOS and treated with immunochemotherapy, those with high *LMO2* expression (defined by a cut-off *LMO2* H-score over 15) exhibited a median overall survival (OS) of 162 months and a 5-year OS of 100%, compared to 132 months and 57%, respectively, for cases with an *LMO2* H-score below 15 (*p* = 0.016), as per patients dying of lymphoma (Figure 4). In Cox regression survival analyses, including the International Prognostic Index (IPI) and *LMO2* H-score, both variables predicted the OS (HR: 1.91, *p* = 0.02 and HR: 0.23, *p* = 0.05, respectively). The 5-year progression-free survival was 58% and 80% (*p* = 0.2), respectively. The survival analyses including patients dying of lymphoma or other causes showed similar results (Figure 4A–C).

## 4. Discussion

In this study, our primary aim was to evaluate *LMO2* mRNA expression in tissue, an area with limited research. While there have been some investigations into *MYC* ISH expression in normal tissue and lymphoma [20,21], the data on *LMO2* and the correlations described here are novel. Although the number of cases studied by GEP is limited, the significant statistical results obtained after comparing the three different scoring systems evaluated between themselves and GEP indicate that *LMO2* ISH is a valid method to evaluate *LMO2* mRNA expression in tissues. In addition, previous studies, undertaken by our own group and others, have obtained similar findings when comparing LMO2 expression and *MYC* FISH results [10,11,22,23,24]. Interestingly, we identified varying levels of *LMO2* expression within the normal GC, and high *LMO2* expression appeared in the LZ of the normal GC, compared with the DZ. This is a feature not captured by IHC and has been shown in other studies evaluating GEP profiles of normal tissues and lymphoma [25,26]. A feasible explanation for the LMO2 IHC and CISH dissociation in the GC may be related to specific *LMO2* and *MYC* functions. As both *MYC* and *LMO2* cause defects in the repair of double-strand DNA breaks (DSB) [27,28], restraining *LMO2* in the DZ may be a useful mechanism to avoid increased DNA damage in *MYC*-exposed replicating cells reentering from the LZ. In the normal GC, the downregulation of *LMO2* is lower than in lymphomas, and the differences in the average lifespan between *LMO2* mRNA and LMO2 protein, particularly in a high proliferation compartment such as the DZ, can be a plausible reason to explain the differences in the sensitivity observed between IHC and CISH in the normal GC. 

We have also observed a remarkable decrease of *LMO2* expression occurring in aLBCL compared with normal expression in the GC, which is lower in tDLBCL from FL and GCB-like DLBCL-NOS than in the other aLBCL with *MYCr*. Moreover, the inverse correlation between normal *LMO2* and *MYC* CISH expression, much more deregulated in aLBCL carrying *MYCr* than *MYC* gains, was quantifiable by using H-scores. Although the precise mechanisms underlying the interaction between LMO2 and MYC are not established, our findings highlight the inverse relationship between them as identified by CISH. The opposing correlation between LMO2 expression and the presence of *MYCr* was already noted in previous studies made by our group [10,11,22,29]. The association between both markers from a protein level standpoint is not so strong because MYC protein overexpression does not capture the presence of *MYCr* as well as the lack of LMO2 protein expression does. We hypothesize that the downregulation of LMO2 may be a survival mechanism for the tumor cells carrying *MYCr*. Because both *MYC* and *LMO2* genes impair the pathways involved in the repair of DSB, the silencing of *LMO2* may be a mechanism to avoid spontaneous apoptosis and higher susceptibility to chemotherapy in cases carrying high levels of DNA damage. We have observed that some *MYCr*/LMO2+ aLBCL have a peculiar clinical behavior, and patients carrying *MYCr*, even with additional *BCL2/BCL6* rearrangements, show very long survival after treatment. We reported this observation in our previous study, and further studies may clarify this observation [10]. Additional mechanisms that may explain the inverse expression between *MYC* and *LMO2* may be related to the deregulation of *E2A/TCF3* in Burkitt lymphoma or *BCL2/BCL6* in DLBCL with *MYCr* or may perhaps be the result of the normal downregulation of *LMO2* mRNA in the DZ of the germinal center in lymphomas that arise in the DZ and show a DZ signature [30,31].

The survival analyses, although performed in a limited number of cases, captured the favorable prognostic impact of high *LMO2* mRNA expression identified in GEP and protein studies [4,5,6,9]. It is known that cells expressing LMO2 are unable to properly repair their genetic material and are more susceptible to being treated with chemotherapy and may be vulnerable to PARP inhibitors as well. This mechanism, elegantly described in DLBCL, may be also relevant in T-cell lymphoblastic leukemia, reinforcing the protective role of LMO2 in hematological neoplasms [9,28,32]. 

## 5. Conclusions

In summary, our study demonstrates a reliable method for identifying *LMO2* mRNA expression using CISH, which effectively captures many of the reported biological features of LMO2. Further studies may increase the understanding of LMO2 biology in aLBCL, as well as in other blood and solid tumors, and may use the present novel technical approach for this purpose to study *LMO2* mRNA expression in tissue.

## Figures and Tables

**Figure 1 cancers-16-02378-f001:**
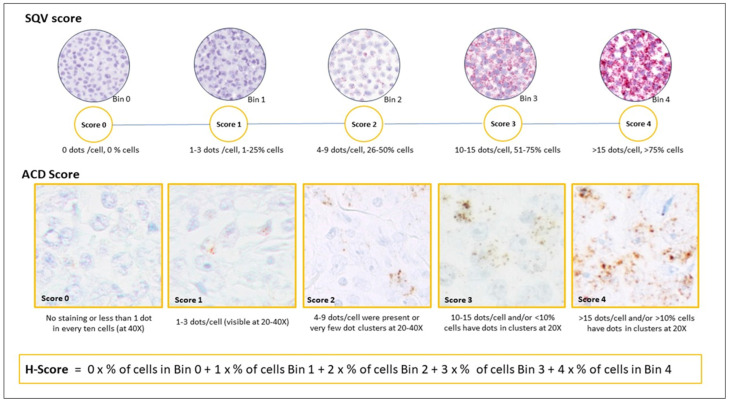
Scoring systems for *LMO2* and *MYC* mRNA CISH analyses (SQV, semiquantitative, ACD, semiquantitative and clusters, H-score, weighted score).

**Figure 2 cancers-16-02378-f002:**
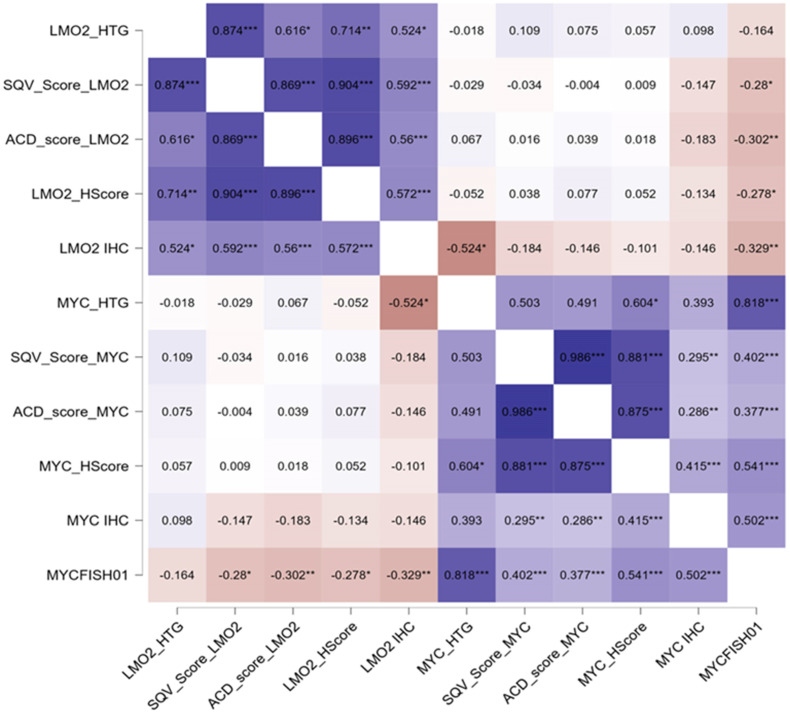
*LMO2* and *MYC* CISH scoring systems (SQV, semiquantitative, ACD, semiquantitative and clusters, H-score, weighted score) correlations with GEP cases (*LMO2*_HTG, *MYC*_HTG), protein expression (LMO2 IHC, MYC IHC), and *MYC* FISH (MYC FISH01). All LMO2-related results showed significant associations between them, indicating the utility of the 3 methods to evaluate *LMO2* CISH. * *p* < 0.05; ** *p* < 0.01; *** *p* < 0.001.

**Figure 3 cancers-16-02378-f003:**
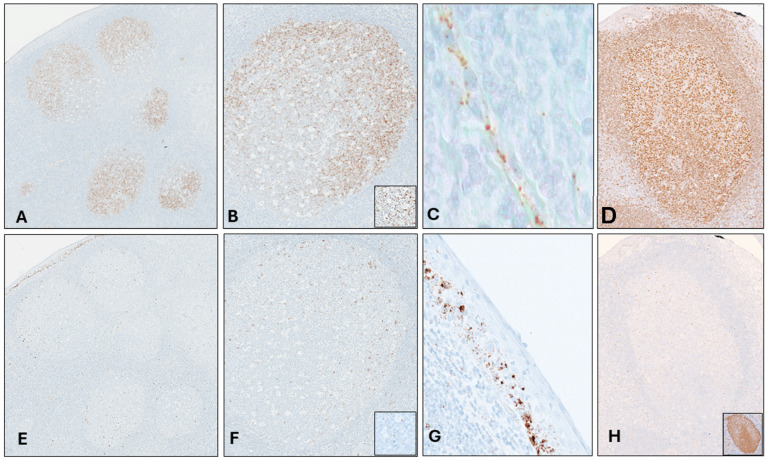
*LMO2* and *MYC* CISH in tonsil. High *LMO2* CISH expression in germinal centers (GC) with higher expression in the light zone ((**A**,**B**), insert); internal control from endothelial cells (**C**); high LMO2 homogeneous IHC expression in the GC (**D**); *MYC* CISH in the GC with high expression in the light zone ((**E**,**F**), insert); internal control from basal epithelial cells (**G**); few MYC protein cells in the light zone of the GC ((**H**); Ki-67, insert).

**Figure 4 cancers-16-02378-f004:**
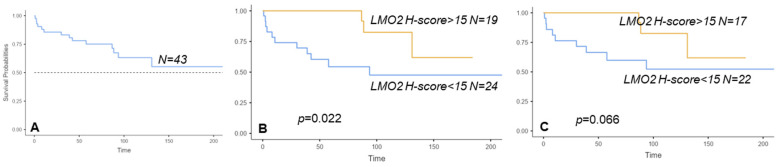
Overall survival (OS) of 43 patients treated with immunochemotherapy censored by death (**A**–**C**) and by death of lymphoma (**D**–**F**). OS analysis excluding 4 patients treated with R-CVP in each category (**C**,**F**).

**Table 1 cancers-16-02378-t001:** Clinicopathological features of aLBCL included in the study.

	All Cases	tDLBCL from MZL	tDLBCL from FL	DLBCL-NOS	HGBCL DH/TH	HGBCL-NOS	BL
Number of cases	82	1	12	56	9	2	2
Age	67 (38–93)	63	65 (47–80)	67 (38–88)	73 (59–93)	53 (53–54)	43 (38–49)
Sex (male:female)	44:38	1:0	7:5	28:28	6:3	1:1	1:1
Nodal	58 (71%)	1 (100%)	12 (100%)	36 (64%)	7 (78%)	2 (100%)	2 (100%)
IPI *							
Low	22/77 (29%)	0/1 (0)	3/10 (30%)	19/54 (35%)	0/8 (0)	0/2 (0)	0/2 (0)
Low-intermediate	16/77 (21%)	0/1 (0)	4/10 (40%)	11/54 (20%)	0/8 (0)	1/2 (50%)	0/2 (0)
High-intermediate	24/77 (31%)	1/1 (100%)	3/10 (30%)	13/54 (24%)	4/8 (50%)	1/2 (50%)	2/2 (100%)
High	15/77 (19%)	0/1 (0)	0/10 (0)	11/54 (20%)	4/8 (50%)	0/2 (0)	0/2 (0)
Immunohistochemistry							
CD10	43 (52%)	0	8 (67%)	25 (45%)	6 (67%)	2 (100%)	2 (100%)
BCL-6	76 (93%)	1 (100%)	11 (92%)	52 (93%)	8 (89%)	2 (100%)	2 (100%)
MUM1/IRF4	52 (63%)	1 (100%)	4 (33%)	43 (78%)	2 (22%)	2 (100%)	0
MYC	34 (41%)	0	5 (42%)	17 (30%)	8 (89%)	2 (100%)	2 (100%)
LMO2	48 (58%)	1 (100%)	10 (83%)	34 (61%)	3 (33%)	0	0
BCL-2	66 (80%)	1 (100%)	11 (92%)	45 (80%)	8 (89%)	1 (50%)	0
COO							
GCB-like	49 (60%)	0	9 (75%)	29 (52%)	8 (89%)	2 (100%)	2 (100%)
Non-GCB-like	33 (40%)	1 (100%)	3 (25%)	27 (48%)	1 (11%)	0	0
FISH							
*MYC* rearranged	20/82 (24%)	0/1 (0)	5/12 (42%)	2/56 (4%)	9/9 (100%)	2/2 (100%)	2/2 (100%)
*BCL-2* rearranged *	20/68 (24%)	0/1 (0)	8/12 (67%)	5/42 (12%)	7/9 (78%)	0/2 (0)	0/2 (0)
*BCL-6* rearranged *	17/61 (28%)	0/1 (0)	5/11 (45%)	9/36 (25%)	3/9 (33%)	0/2 (0)	0/2 (0)

* IPI was available in 77 cases, and FISH for *BCL2* and *BCL6* in 68 and 61 cases, respectively.

**Table 2 cancers-16-02378-t002:** Median and quartiles (Q1, Q3) of *LMO2* and *MYC* mRNA ISH H-score values in normal tonsil and aLBCL.

	*LMO2* H-Score	*MYC* H-Score
Control tonsil		
GC dark zone	115 (110, 120)	40 (30, 40)
GC light zone	260 (250, 275)	120 (115, 125)
Mantle/marginal zones	30 (30, 40)	50 (30, 50)
Interfollicular areas	30 (20, 40)	50 (40, 55)
All cases (*N* = 82)	15 (5, 90)	30 (8.75, 90)
tDLBCL from MZL (*N* = 1)	110	140
tDLBCL from FL (*N* = 12)	97.5 (42.5, 127.5)	75 (56.25, 162.5)
DLBCL-NOS (*N* = 56)	15 (5, 72.5)	25 (5, 47.5)
GCB-like (*N* = 29)	40 (7.5, 82.5)	20 (5, 52.5)
Non-GCB-like (*N* = 27)	10 (5, 50)	30 (5, 50)
HGBCL DH/TH (*N* = 9)	5 (5, 12.5)	100 (32.5, 177)
HGBCL-NOS (*N* = 2)	5	190
BL (*N* = 2)	7.5	155

**Table 3 cancers-16-02378-t003:** Characteristics of the whole series classified according *MYC* rearrangements.

Diagnosis	*MYC* Rearranged (*N* = 20)	*MYC* Non-Rearranged (*N* = 62)
tDLBCL	5/20 (25%)	8/62 (13%)
DLBCL-NOS	2/20 (10%)	54/62 (87%)
HGBCL-NOS	2/20 (10%)	-
HGBCL DH/TH	9/20 (45%)	-
BL	2/20 (10%)	-
Nodal/Extranodal	14 (70%)/6 (30%)	44 (71%)/18 (29%)
CD10 expression	14/20 (70%)	29/62 (47%)
GCB-like/Non-GCB-like *	16 (80%)/4 (20%)	33 (53%)/29 (47%)
*LMO2* GEP, median (Q1;Q3)	1301 (638; 2848.5)	1679.5 (1056; 5207.75)
*LMO2* H-score, median (Q1;Q3)	5 (5; 37.5)	35 (5; 102.5)
LMO2 protein expression	6/20 (30%)	42/62 (68%)
*MYC* GEP, median (Q1;Q3)	5171 (3638; 6879)	1025.5 (804.25; 1688)
*MYC* H-score, median (Q1;Q3)	157.5 (72.5; 190)	30 (5; 56.25)
MYC protein expression	17/20 (85%)	17/62 (27%)

* Classified following Hans algorithm.

## Data Availability

The datasets generated during and/or analyzed during the current study are available from the corresponding author upon reasonable request.

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
