# Peer review of "Chromogenic LMO2 mRNA ISH Expression Correlates with LMO2 Protein and Gene Expression and Captures Their Survival Impact in Diffuse Large B-Cell Lymphoma, NOS"

_cancers, 2024, doi:10.3390/cancers16132378_

Round 1

Reviewer 1 Report

Comments and Suggestions for Authors

1. The sample sizes for some of the comparisons are relatively small, which may limit the generalizability of the findings. Please address how this limitation might affect the reliability of the findings.

2. Based on the current findings of this study, there is an inverse correlation between LMO2 and MYC expression, the underlying mechanisms remain unclear as cited by the authors. Detailed exploration or hypothesis on possible mechanisms could enhance the discussion. Also, the study could also be enhanced by highlighting whether this inverse relationship between LM02 expression and MYC r is carried over to 11p13 (LMO2 mapped to 11p13) alterations and MYC r.

 3. The discussion could be strengthen by comparing their findings more extensively with other studies on LMO2 and MYC expressions and  identifying any consistencies or discrepancies.

 4. Some parts of the discussion could benefit from further explanation. For example, the detailed relationship between LMO2 expression levels in LZ and DZ and their clinical implications could be elaborated further.

5. Cytogenetics was performed on these cases, however, the specific cytogenetic findings are not included. Including this data could provide valuable context for future work. Therefore, please include the cytogenetic findings for these cases in a supplementary table. Providing this additional information would greatly benefit both the readers and researchers by offering a more comprehensive picture of the genetic landscape associated with LMO2 expression in your cohort.

6. Similarly to comment #2, on the 15 cases that were analyzed by GEP, the authors could discuss or provide a comparison between LMO2 expression levels and the status of chromosome 11p13 (LM02 mapped to this region).

Author Response

Comments and Suggestions for Authors

  1. The sample sizes for some of the comparisons are relatively small, which may limit the generalizability of the findings. Please address how this limitation might affect the reliability of the findings.

We agree with the reviewer in that the RNA transcriptome sample is relatively small. However, GEP and LMO2 protein results are similar as obtained in our previous study (Colomo L et al, Am J Surg Pathol. 2017;41:877-886) using a larger series of patients. On the other hand, we have obtained similar results and strong statistical correlations after comparing gene expression profiling (GEP) and RNA CISH by the 3 different methods analyzed. Thus, based on these results, we decided to study further our series of patients. We have added this information in the first paragraph of the Discussion. 

  1. Based on the current findings of this study, there is an inverse correlation between LMO2 and MYC expression, the underlying mechanisms remain unclear as cited by the authors. Detailed exploration or hypothesis on possible mechanisms could enhance the discussion. Also, the study could also be enhanced by highlighting whether this inverse relationship between LM02 expression and MYCr is carried over to 11p13 (LMO2 mapped to 11p13) alterations and MYCr.

In our previous studies we observed an inverse correlation between LMO2 expression and the presence of MYC rearrangements. The correlation between both markers at the protein level is not so strong because MYC protein overexpression does not capture the presence of MYCr as well as the lack of LMO2 protein expression. We hypothesize that the downregulation of LMO2 may be a survival mechanism of the tumor cells carrying MYCr.

As both MYC and LMO2 genes cause defects in the repair of double-strand DNA breaks (DSB) (Z. Jin, Cell 2006;281:14446-14456; S. Parvin, Cell 2019;36:237-249), silencing LMO2 may be a mechanism to avoid spontaneous apoptosis and higher susceptibility to chemotherapy in such cases. We have observed that some MYCr/LMO2+ cases have a peculiar clinical behavior, and patients carrying MYCr, even with additional BCL2/BCL6 rearrangements, are very long survivors. We reported this observation in our previous study (I. Vazquez, Cancers 2020;12:884) and we are trying to expand our series evaluating more cases. Additional mechanisms may be related to the deregulation of E2A/TCF3 in Burkitt lymphoma or BCL2/BCL6 in DLBCL with MYCr, or perhaps may be the result of the normal downregulation of LMO2 mRNA in the dark zone (DZ) of the germinal center that we have observed in the present study, increased in lymphomas that arise in the DZ and show a DZ signature (W. Alduaij, Blood 2023;141:2493-2507). We have added this information in the reviewed version and highlighted it in yellow.    

Regarding 11p13 alterations, there is few information regarding LMO2 gene. The study of Durnik et al. (DK Durnik, Am J Clin Pathol 2010;134:278-281) did not show LMO2 translocations in 101 DLBCL studied. The authors did not identify LMO2 structural alterations using a non-commercial break-apart LMO2 probe produced using BACs. We also produced LMO2 probes using BACs to test the numeric changes of LMO2. In 47 cases studied we did not find robust associations between the numerical alterations of LMO2 and LMO2 protein expression or MYCr (data not published).

  1. The discussion could be strengthen by comparing their findings more extensively with other studies on LMO2 and MYC expressions and  identifying any consistencies or discrepancies.

We have added other and our studies describing the inverse association between LMO2 expression and MYCr. The new papers have been included in the Bibliography section.

  1. Some parts of the discussion could benefit from further explanation. For example, the detailed relationship between LMO2 expression levels in LZ and DZ and their clinical implications could be elaborated further.

In the present study we have identified varying levels of LMO2 expression within the normal GC and high LMO2 expression appeared in the LZ of the normal GC, compared with the DZ. This is a feature not captured by IHC and has been shown in other studies evaluating GEP profiles of normal tissues and lymphoma (R. Massoni-Badosa, Immunity 2023; P. Milpied, Nat Immunol 2018). May be this is one of the mechanisms explaining the dissociation of MYCr and LMO2 protein expression in aLBCL. May be the physiological downregulation of LMO2 in the DZ allow the MYC-induced replicating cells reentering from the LZ avoid the accumulation of too much DSB. This information has been added in the Discussion.  

  1. Cytogenetics was performed on these cases, however, the specific cytogenetic findings are not included. Including this data could provide valuable context for future work. Therefore, please include the cytogenetic findings for these cases in a supplementary table. Providing this additional information would greatly benefit both the readers and researchers by offering a more comprehensive picture of the genetic landscape associated with LMO2 expression in your cohort.

We have added an additional table in the manuscript (Table 3) distinguishing the features of MYCr and non-MYCr cases, in order to gain useful information.

  1. Similarly to comment #2, on the 15 cases that were analyzed by GEP, the authors could discuss or provide a comparison between LMO2 expression levels and the status of chromosome 11p13 (LM02 mapped to this region).

We are sorry, but we unfortunately do not have LMO2 FISH results of GEP studied cases.

Reviewer 2 Report

Comments and Suggestions for Authors

Chromogenic LMO2 mRNA ISH Expression Correlates with 2 LMO2 Protein and Gene Expression and Captures their Survival Impact in Diffuse Large B-Cell Lymphoma, NOS.

Natalia Papaleo and colleagues.

Thank you for asking me to review the above MS.  I must apologize to the authors for the slight delay in my review.

LMO2 is a LIM domain containing transcription factor that has multiple and critical roles in hemopoiesis.  It is expressed at high levels within the normal germinal center (GC) and also in some GC derived B cell lymphomas.  High level expression of LMO2 has previously been associated with a favorable outcome to immunochemotherapy.  These data have been well established using both RNA-based methods and immunohistochemistry.  LMO2 is not, as far as I am aware, a component of modern prognostic models for DLBCL.

This MS seeks to determine the possible clinical utility of an in situ hybridization assay for LMO2 RNA in aggressive lymphoma arguing that such methods allow preservation of tissue architecture.  Whilst this is undoubtedly the case, the clinical necessity for these experiments is not clear to me given the availability of perfectly good antibodies for IHC.  What is gained clinically by performing ISH?

Where the data are potentially more interesting is where there are apparent discrepancies between RNA and protein expression eg Figure 3 showing high expression of LMO2 RNA in the light zone but comparable levels of LMO2 protein throughout the GC.  What is the basis for this discrepancy please?  Have the RNA differences been detected in other datasets?  (not sure that ref 21 is the best in this context – please see for example DOI:https://doi.org/10.1016/j.immuni.2024.01.006

Importantly do LMO2 RNA high/LMO2 protein low cases of lymphoma behave differently?

Otherwise, there are very few new data in the MS.

Other criticisms

a)     Please amend title.  LMO2 mRNA – LMO2 should be in italics please.  Also, please spell out ISH before contraction.

b)     An important reference appears to be missing, which confirms many of the data reported in this MS.  doi: 10.1080/10428194.2021.1927020

Author Response

Reviewer 2

Comments and Suggestions for Authors

Chromogenic LMO2 mRNA ISH Expression Correlates with 2 LMO2 Protein and Gene Expression and Captures their Survival Impact in Diffuse Large B-Cell Lymphoma, NOS.

Natalia Papaleo and colleagues.

Thank you for asking me to review the above MS.  I must apologize to the authors for the slight delay in my review.

LMO2 is a LIM domain containing transcription factor that has multiple and critical roles in hemopoiesis.  It is expressed at high levels within the normal germinal center (GC) and also in some GC derived B cell lymphomas.  High level expression of LMO2 has previously been associated with a favorable outcome to immunochemotherapy.  These data have been well established using both RNA-based methods and immunohistochemistry.  LMO2 is not, as far as I am aware, a component of modern prognostic models for DLBCL.

We agree with the reviewer. Modern prognostic models of DLBCL are mostly based in cytogenetic alterations and mutational profiles of the tumors. Alterations of 11p13, the locus that maps LMO2, are uncommon in aLBCL and, to the best of our knowledge, there is only one study focusing in LMO2 translocations (DK Durnik, Am J Clin Pathol 2010;134:278-281), that did not find structural alterations of the gene in a large series of DLBCL. We have produced LMO2 probes using BACs to test numeric changes of LMO2. In 47 cases studied we did not find robust associations between the numerical alterations of LMO2 and LMO2 protein expression or MYCr (data not published). On the other hand, mutational studies do not report to date clinically significant alterations of LMO2. Contrarily, GEP-based studies have shown the clinical prognostic impact of LMO2 in aLBCL.

This MS seeks to determine the possible clinical utility of an in situ hybridization assay for LMO2 RNA in aggressive lymphoma arguing that such methods allow preservation of tissue architecture.  Whilst this is undoubtedly the case, the clinical necessity for these experiments is not clear to me given the availability of perfectly good antibodies for IHC.  What is gained clinically by performing ISH?

We agree with the reviewer. There are some antibodies of LMO2 that have shown proven utility in clinical practice, and they can be applied in the work-up of aggressive lymphomas. However, there is fewer information available in other hematological neoplasms except for T-lymphoblastic leukemia. We think that LMO2 ISH may be a useful tool to study other lymphoma types showing variable expression of the protein, as occurs in low grade follicular lymphoma (Y. Natkunam, Blood 2007 109: 1636-1642; C. Agostinelli, Histopathology 2012, 61, 33–46) and other solid tumors showing cytoplasmic relocation of the protein (C. Agostinelli, Histopathology 2012, 61, 33–46), as well as studies evaluating the functions of the gene.

In addition, we think that LMO2 ISH may be a useful tool to study the cases with the unexpected pattern of CD10+/MYCr/LMO2 expression. We have observed that some of these cases have a peculiar behavior, and patients carrying MYCr, even with additional BCL2/BCL6 rearrangements, are very long survivors. We reported this observation in our previous study (I. Vazquez, Cancers 2020;12:884) and we are expanding our series.

On the other hand, our observation about the dissociation on LMO2 protein and LMO2 CISH in the light zone of the germinal center may be a plausible explanation of the downregulation of LMO2 in MYCr/CD10+ aLBCL. This information has been added in the Discussion. 

Where the data are potentially more interesting is where there are apparent discrepancies between RNA and protein expression eg Figure 3 showing high expression of LMO2 RNA in the light zone but comparable levels of LMO2 protein throughout the GC.  What is the basis for this discrepancy please?  Have the RNA differences been detected in other datasets?  (not sure that ref 21 is the best in this context – please see for example DOI:https://doi.org/10.1016/j.immuni.2024.01.006

We thank the reviewer for the reference. We are adding it in the manuscript.  

We do not know the precise reasons for the discrepancy between LMO2 mRNA CISH expression and LMO2 protein expression in the LZ and DZ of the germinal center. A feasible explanation of the LMO2 IHC and CISH dissociation in the GC may be related to the levels of LMO2 and MYC we have identified using the H-score in normal tonsil and lymphomas described in the present study. As both MYC and LMO2 cause defects in the repair of double-strand DNA breaks (DSB) (Z. Jin, Cell 2006;281:14446-14456; S. Parvin, Cell 2019;36:237-249), silencing LMO2 in the DZ may be a useful mechanism to avoid increased DNA damage in MYC-exposed replicating cells reentering from the LZ. In the normal GC the downregulation of LMO2 is lower than in lymphomas, and the differences in the average lifespan between LMO2 mRNA and LMO2 protein in a proliferation compartment such as the DZ can be a plausible reason to explain the discrepancy between IHC and CISH in normal GC. In lymphomas, the higher decrease of LMO2, the architectural alterations of the tissue, and the higher levels of MYC in MYCr aLBCL allow to identify more evidently LMO2 downregulation by both methods. We have added this explanation in the Discussion.    

Importantly do LMO2 RNA high/LMO2 protein low cases of lymphoma behave differently?

We have identified 5 cases in quartiles Q3 and Q4 showing LMO2 mRNA high/LMO2 negative protein expression in this series. They are 2 tDLBCL (cases #1 and #2, 1 tMZL non-MYCr and 1 tFL with MYCr; LMO2 H-scores 50 and 60, respectively), 2 DLBCL (cases #3 and #4, both non-MYCr; LMO2 H-scores 40 and 50), and 1 HGBCL MYC/BCL6 rearranged (case #5, LMO2 H-score 100). Patient #1 was 64 y-o man that died with no evidence of disease after immunochemotherapy due to pneumonia, after a follow-up of 10 months; patients #2, #3 and #4 are alive and in CR after immunochemotherapy, and a follow-up of 111 months, 166 months, and 181 months, respectively. Patient #5 was a 82 y-o man that did not receive treatment because the disease presented at advanced stage, and died 5 days after the diagnosis due to broncoaspiration.

The behavior of these patients is closer to those with high LMO2 protein expression, reinforcing the value of our LMO2 CISH results in clinical practice. This is an interesting observation that should be further explored. 

Otherwise, there are very few new data in the MS.

In this manuscript we wanted to test the utility of a new tool, that we think we are demonstrating as useful. We consider that the fact that our generated data reproduced the other approaches is consistent and useful and do not undervalue the present study, particularly thinking about the possibilities of the usage of the new approach.    

Other criticisms

  1. a)     Please amend title.  LMO2 mRNA – LMO2should be in italics please.  Also, please spell out ISH before contraction.

We have corrected LMO2 in the title. We are spelling ISH in the abstract. The reason is that we think that ISH is an understandable contraction for pathologists’ community and the title is already too long to increase it more.   

  1. b)     An important reference appears to be missing, which confirms many of the data reported in this MS.  doi: 10.1080/10428194.2021.1927020

The reference has been included.

Reviewer 3 Report

Comments and Suggestions for Authors

The study entitled „Chromogenic LMO2 mRNA ISH Expression Correlates with LMO2 Protein and Gene Expression and Captures their Survival Impact in Diffuse Large B-Cell Lymphoma, NOS” is interesting and focuses on novel method of new marker assessment LMO2 – transcription factor LIM-domain only. The study is well written and prepared.

However I have some questions that need to be addressed. At some point the information is missing which cases were included into GEP analysis?

Are the DLBCL cases/samples derived from tonsils?

What was the origin of normal, non-neoplastic tonsil?

Figure 1 – was this scoring system developed for tonsilar tissue?

In Figure 3 – letter “D” is missing

Line 219: what is meant by “(G); Few MYC protein cells”?

Author Response

Comments and Suggestions for Authors

The study entitled „Chromogenic LMO2 mRNA ISH Expression Correlates with LMO2 Protein and Gene Expression and Captures their Survival Impact in Diffuse Large B-Cell Lymphoma, NOS” is interesting and focuses on novel method of new marker assessment LMO2 – transcription factor LIM-domain only. The study is well written and prepared.

However I have some questions that need to be addressed. At some point the information is missing which cases were included into GEP analysis?

We thsnk the reviewer thiscomment. We have updated this information in Material and Methods section and highlighted in yellow.

Are the DLBCL cases/samples derived from tonsils?

No. This series includes 82 cases, 58 (71%) of them obtained from the lymph nodes, and 24 (29%) from extranodal sites, and there were no samples arising specifically from the tonsil.

What was the origin of normal, non-neoplastic tonsil?

The non-neoplastic tonsil was derived from a young patient with tonsillar hyperplasia with no active EBV or other viral or bacterial infections.

Figure 1 – was this scoring system developed for tonsilar tissue?

Not specifically. We followed the scoring systems proposed by the manufacturer. Because there were some differences between them, we decided to test the 3 scoring systems proposed. These scores were not developed only for tonsillar tissue and several manuscripts have already applied them to different tissue types and tumors.

In Figure 3 – letter “D” is missing

Thank you. We have corrected the missing letter.

Line 219: what is meant by “(G); Few MYC protein cells”?

It means that there are few cells expressing MYC protein in the light zone of the germinal center. This is physiological, and such cells recirculate to the dark zone to experience additional rounds of proliferation and further acquisition of IG somatic mutations.

Round 2

Reviewer 2 Report

Comments and Suggestions for Authors

Thank you for responding so fully to the points raised. I have no additional comments.